# “It Doesn’t Cure, but It Protects”: COVID-19 Vaccines through the Eyes of Children and Their Parents

**DOI:** 10.3390/vaccines11081305

**Published:** 2023-07-31

**Authors:** Candice Groenewald, Dane Isaacs, Mafanato Maluleka

**Affiliations:** 1Human and Social Capabilities Division, Human Sciences Research Council, Durban 4001, South Africa; mmaluleka@hsrc.ac.za; 2Department of Psychology, Rhodes University, Makhanda 6139, South Africa; 3South African Research Ethics Training Initiative, University of KwaZulu-Natal, Pietermaritzburg 3209, South Africa; 4DSI-NRF Centre of Excellence in Human Development, University of the Witwatersrand, Johannesburg 2000, South Africa; 5Desmond Tutu Centre for Religion and Social Justice, University of the Western Cape, Cape Town 7535, South Africa; isaacsdane@gmail.com

**Keywords:** adolescents, intergenerational knowledge, pandemic, South Africa, vaccine acceptability

## Abstract

Recently, studies have examined COVID-19 vaccine acceptance and/or hesitancy amongst adult populations across the globe. However, there is a paucity of literature illustrating children’s voices in vaccination debates. This article draws on qualitative data collected via a mixed-methods study that explored South Africans’ experiences during the COVID-19 pandemic between 2020 and 2021. Interviews were conducted with a purposive sample (*N* = 29) of children (>18 years) and their parents regarding their initial perspectives on COVID-19 vaccines. Given the dyadic nature of our study, we explored the intergenerational influence that parents’ perspectives had on children’s vaccine acceptability and the role that vaccine literacy, or lack thereof, played in vaccine decision making. Findings showed a great level of vaccine acceptability among children and parents, where many placed hopes in the vaccines to promote societal health and wellbeing. Intergenerational transfer of perspectives was observed where children’s willingness to receive a vaccine was intrinsically linked to their parents’ vaccine acceptability. Some participants also expressed concerns about COVID-19 vaccines, related to misinformation, mistrust, and limited vaccine literacy. We discuss the findings as they relate to vaccine and health literacy, also considering the prospective implications of this work as we enter the “recovery” period of the pandemic.

## 1. Introduction

The COVID-19 pandemic significantly disrupted the lives of children, adolescents, adults, and families, who were required to be hypervigilant about their social movements and engagements, enforced by national regulations such as “lockdowns” or “stay at home orders”. People were also compelled to become more health literate by rapidly learning about COVID-19 and observing protective health protocols such as mask-wearing, social distancing, and sanitizing. These unexpected disruptions, along with the deathly nature of COVID-19, produced significant distress amongst younger and older populations, where feelings of anxiety, sadness, stress, and fear have been reported [1,2].

In addition to coping with the constantly evolving stressful reality of the pandemic, people were expected to make decisions about whether they wanted to receive a COVID-19 vaccine or not [3]. Global vaccination was considered a priority to reduce the spread of the coronavirus and to achieve herd immunity [4,5]. Yet, hesitancy to receive a COVID-19 vaccine was an evolving threat to public health globally [6,7]; it also received significant attention in South Africa [3,8]. COVID-19 vaccine hesitancy has been closely related to subjective and societal perceptions of vaccines and doubts in the efficacy of these vaccines [8,9].

Along with vaccinations for adults, South Africa (as other countries worldwide) extended COVID-19 vaccination to children older than 12 years; a cohort that does not require parental consent to receive a vaccine. This aligns with the Children’s Act, which allows minors to access medical treatment without parental consent under specific guidelines [10]. It also models international laws (e.g., the Pennsylvania law) that authorize minors’ consent for the treatment of reportable diseases, including COVID-19 [11]. International studies show mixed results on children’s (<18 years) vaccine acceptability, where increased acceptance was reported amongst children in China [12] and the US [13], while lower acceptability was found in Hong Kong [14] and England [15]. Uptake of vaccines amongst minors is also generally dependent on parents’ or caregivers’ acceptability of those vaccines, which is further associated with their subjective intentions to vaccinate [16].

In this article, we describe South African parents’ and their children’s perceptions of COVID-19 vaccines, focusing also on the associations that exist between parents’ perceptions and those of their children, to explicate the intergenerational transfer of vaccine acceptability.

## 2. Materials and Methods

This article formed part of a longitudinal study, using a mixed-methods design, that explored the impact of the COVID-19 pandemic on family life amongst a purposive sample of South African children and adults (*N* = 45). The study was conducted over a one-year period, between January and December 2021, where qualitative and quantitative data were collected at three time points. Participants included various cohorts such as children, adolescents, adults (including parents and/or caregivers and students), and older persons (60+ years). This article drew on data collected through the qualitative component of the study with those participants who were parents and their children (*N* = 29). COREQ principles were applied to describe the steps followed in the qualitative component (see Appendix A for COREQ checklist) [17].

Given the physical distancing restrictions imposed during the lockdown in South Africa, a digital snowball approach [18] was adopted to recruit participants via social media platforms such as Facebook, Instagram, and WhatsApp (an instant messaging system) statuses. Figure 1 outlines the digital snowball approach followed in this study. We also invited participants who participated in a sister study called Life During Lockdown to participate in the current study (see, for example, [19]). Minors (17 years and younger) were recruited via their parents or caregivers who provided consent for their participation and the use of their transcripts. Participant assent was obtained from all minors who participated in the study and all adult participants also provided consent for their participation and the use of transcripts.

Narratives from twenty-nine (*N* = 29) participants who resided across six provinces in South Africa, namely Eastern Cape, Gauteng, KwaZulu-Natal, Limpopo, North-West Province, and the Western Cape, are included in this paper. This included sixteen minors, of whom six were children (6 to 11 years) and ten were adolescents (12–17 years) (see Table 1). Amongst the adult participants, thirteen were parents and the majority were mothers (n = 10). Table 2 displays the relationships between the parents and children who participated in the study. To ensure anonymity and confidentiality, participant codes were assigned to each family group, ensuring that we linked adult participants with their respective children or adolescents within each family. The following conventions were used: family group (e.g., PS1 or PCH1); participant number (e.g., 1); gender (e.g., f = female; m = male).

To abide by lockdown regulations, interviews were conducted telephonically, while researchers also engaged with participants via WhatsApp (an instant messaging application). Interviews were facilitated by a semi-structured interview guide that explored a range of issues, with primary interest in exploring participants’ lived experiences during the pandemic (see interview guide in Appendix A). In this article, we focused on the participants’ perspectives related to COVID-19 vaccines. Interview guides were developed through an iterative consultative process by two researchers (Authors 1 and 3), with the intention of using simple questions to capture participants’ thoughts and experiences of events as they unfolded during the COVID-19 lockdown. Data were collected by researchers who had experience in conducting qualitative interviews and who were proficient in various South African vernacular languages. Interviews were generally conducted in English or, where required, a vernacular language such as Zulu or Sotho as requested by participants. The duration of the interviews was between 30 min and 1 hr. All interviews were audio recorded with permission from the participants and their parents and were subsequently transcribed and translated (where relevant). Authors 1 and 3 conducted a thematic analysis [20] on the transcripts, supported by Atlas ti (version 9) software. Open inductive coding was initially employed by these two authors, followed by “an iterative process of formulating categories and exploring relationships between categories” [21]. Author 2, in consultation with Author 1, conducted a secondary review of the codes and themes that emerged during the initial coding, verifying the data analysis processes.

### 2.1. Establishing Trustworthiness of Data

Given the complexities embedded in qualitative studies, researchers are required to engage with the notion of trustworthiness within research. Trustworthiness entails the “quality, authenticity, and truthfulness of findings in qualitative research” [22]. To illustrate trustworthiness, we applied Lincoln and Guba’s (1986) [23] four-pronged criteria associated with (1) credibility, (2) transferability, (3) dependability, and (4) confirmability.

Various efforts to ensure credibility, i.e., confidence that the findings were “true” reflections of participants’ accounts, were implemented throughout the study. A primary factor that contributed to the credibility of our study was involving researchers who spoke vernacular languages to conduct the interviews. Participants could thus engage in a language they were most comfortable with or proficient in, or could switch as needed, avoiding any pressure to answer “correctly”. To further avoid (as far as possible) response bias, researchers established rapport with study participants over a three-to-four-week period prior to conducting the interviews. This was conducted through engagements via WhatsApp (an instant messaging application), where researchers were available to answer any questions that participants had. All interviewers had a minimum of three years’ experience in qualitative data collection, including telephonic data collection through our sister study Life During Lockdown. Researchers were not only trained in the data collection tools used in the current study, but also had the opportunity to engage with the questions and amend them to ensure they were contextually appropriate and used relatable language for adults and children. Moreover, debriefing meetings were held on a rolling basis to discuss any issues and to identify strategies to overcome challenges. The same researchers who collected the data also transcribed and translated (where needed) the audio recordings. However, to avoid biases during transcriptions or translations, audios were not transcribed by the same person who conducted the interview. Still, researchers were able to consult each other during the capturing phase to ensure that documents were transcribed adequately; quality control of all transcripts was conducted by the project manager (who was proficient in various languages). Throughout the analysis phase, interviewers were consulted to ensure an appropriate interpretation of the findings.

Transferability refers to the replicability of the findings for different contexts [24]. Sample adequacy and data saturation are considered important components of transferability. Morse et al. [25] described sample adequacy as the “participants who best represent or have knowledge of the topic”. In the current study, we purposively recruited parents and their children to explore how families (as units and as individual members) experienced all components of the COVID-19 pandemic. Data saturation was evident in the groundedness or transferability of the codes within the dataset (as the results showed). Further, we related our findings to the broader literature and, in this way, attended to the transferability of our findings to other contexts. We also noted the limitations of the study, as shown later in this paper.

Dependability or reliability of data was achieved through the iterative coding approach followed in this paper, as explained in Section 2. All codes and themes, with associated extracts, were discussed and verified by all authors; interviewers were also consulted to ensure appropriateness of interpretation. Codes and sub-themes presented in the results were also substantiated by exemplary extracts, as narrated by the participants.

Lastly, confirmability is often associated with researchers’ reflexivity and discussions pertaining to the topic of interest [22,24]. As defined by Cypress (2017, p. 259) [22], reflexivity entails “actively engag[ing] in critical self-reflection about their potential biases and predispositions that they [researchers] bring to the qualitative study. Through reflexivity, researchers become more self-aware and monitor and attempt to control their biases”.

In the current study, this was especially important, as the researchers (including authors and interviewers) were experiencing the same events as the study participants, i.e., the COVID-19 pandemic and expectations to receive a vaccine. Thus, to encourage self-awareness throughout the study, meetings were held with interviewers to discuss the importance of remaining neutral during interviews and to avoid asking leading or probing questions. Researchers were thus asked to remain aware of their positions on vaccinations during the interviews and to use standard responses should participants ask their opinions on whether they should vaccinate or not. Interviewers avoided offering any recommendations with regards to vaccinations. The decision to be vaccinated was solely the decision of the participants. Debriefing meetings also facilitated reflexivity, where researchers could discuss their experiences during interviews and their own views pertaining to the research questions posed to participants. Moreover, the study lead, Author 1, documented her perspectives on COVID-19 vaccinations using autoethnography and subsequently published an article on these reflections [3].

### 2.2. Ethics

Ethical clearance to conduct the study was provided by the Human Sciences Research Council (HSRC) Research Ethics Committee (REC) (approval number REC 2/21/09/16). All participants were asked to provide audio-recorded verbal assent before participating in the interviews. Consent to participate in the full study (surveys and interviews) was provided through a link that was shared digitally with participants. Parental consent was also obtained. This included consent to use data generated from the interviews in publications.

## 3. Results

The findings are presented according to four overarching themes. Extracts are used to illustrate themes, following Groenewald and Bhana’s (2015) conventions [26]. Square brackets “[]” contain material provided by the authors for clarification. Ellipsis points “(…)” indicate that the participants’ thoughts have trailed off; uppercase letters are used to illustrate emphasis. A pause is illustrated by “(.),”; interruptions are indicated by “ = ,”;[…] indicates a break in the conversation. Additional extracts are found in Appendix A.

### 3.1. Parents’ Perceptions of COVID-19 Vaccines

The majority of the parents indicated a willingness to receive a COVID-19 vaccine (n = 8). Initially, parents showed both excitement and hesitancy, signifying the tensions between the hope and uncertainty that parents felt towards these vaccines. Amongst parents who expressed excitement towards the vaccines, many felt optimistic that the vaccines would protect them and their families from severe COVID-19 infection. For example, when asked about how they initially felt when they heard about COVID-19 vaccines, parents explained:

“I was happy because it [COVID-19 vaccine] is here to help us. It doesn’t cure, but it protects”(PCH1-1 1)

“I felt okay. If the vaccine is available, and it works then its fine because everything will go back to normal life. Because things are not the same where you can now conduct business the way you used to. And things will be back to partly normal. I think it is a good move; it is a good thing. I feel good!”(PST1-2)

In addition to viewing the vaccines as protective, the narratives also revealed participants’ hopes that the COVID-19 vaccines would facilitate a return to “normal life”. At the time of the interviews, South Africa was experiencing a nationwide lockdown and thus vaccines were considered a pathway towards “normal life”. However, some participants also raised concerns about the vaccines, questioning the effectiveness and trustworthiness of the vaccines.

“My worries right now [are] there is no vaccine that was tested and found to be effective.”(PST1-1)

“My concern is when will the vaccine reach us the community people? Because sometimes it might not even arrive, or they might want us to even pay for it. And how will people react to it physically and is it the real vaccine that people are waiting for?”(PS3-1)

Reflected in the narratives were participants’ concerns about the safety of COVID-19 vaccines, especially considering the challenges South Africa faced in securing and distributing vaccines during the early vaccination period in 2020. This was best reflected in the second extract above, while another participant explained: “I am just worried about the corrupt people who might manufacture fake vaccines and sell them to enrich themselves. […] There is a lot of corruption…” (PC1-3).

Linked to concerns of safety, participants also described worries related to the potential side effects of COVID-19 vaccines. Given that COVID-19 vaccination was a relatively new approach at the time of the interviews, participants were unsure about what to expect after COVID-19 vaccination. For example: 

“Yes, I feel like there are aftereffects. I do not feel like it is right for all of us to take the vaccine while we do not have symptoms of corona. We might go but after getting the vaccine they might then say there are aftereffects, so I feel like can it show its true colors if it has side effects or not. So, I do not see a need for us to get the vaccine while we do not know the after and before effects of it.”(PS2-1)

Furthermore, a few participants expressed that they had no immediate concerns pertaining to COVID-19 vaccines. This lack of concern was generally related to not having heard any negative news associated with COVID-19 vaccinations as explained below.

“Uh… I don’t really have worries about the vaccines. I was at first [worried], when people were saying vaccines have effects of changing the genes… but I don’t have any concerns about the vaccine now. At first the vaccine was on trial and obviously some people might have reacted to its effects because we all react differently to vaccines. Yes… some people don’t develop side effects. It is not really a major concern for me.”(PC3-1)

### 3.2. Parents Perspectives on COVID-19 Vaccination for Children

Parents were asked if they would be willing to allow their children to receive a COVID-19 vaccine. Parents who indicated that they would be willing to vaccinate themselves also expressed willingness for their child to be vaccinated. Parents’ acceptability of COVID-19 vaccines for children was influenced by different factors. Amongst parents who indicated that they would allow their children to receive a vaccine (n = 10), many (n = 7) cited that vaccines were beneficial to keep children safe from severe COVID-19 infection.

“Yes, I will take him there. He is at a risk since he goes to school with different children who some of them might be infected. I will allow it so that he will be protected against COVID.”(PCH1-1 1)

Two of these parents also justified their decision by reflecting on positive experiences with other vaccination processes as displayed below.

“Yes! I would let them because I have already explained to them that it is like when you went for immunization. I explained to them that they immunized for polio… German measles… Just like before when a child had immunized for German measles, they would tell the child to isolate because it is not 100% that when you have immunized then you would not infect others. It just minimizes the risks of getting infected, so I would advise them to for it. My children also do not have a problem taking the vaccine.” (PC1-1)

“Yes! I will encourage them to take it and also explain to them how it works. I have already explained to them about it anyway. I even show them the vaccination scars that I still have and will tell them, ‘Look, I still have the old scars of vaccinations I received in school back in the days.’ So, vaccine is a good thing.” (PC1-3)

Similarly, parents who were unwilling to vaccinate their children or were skeptical about COVID-19 vaccines for children (n = 4), referred to having bad previous experiences with vaccines. One parent explained: “My child’s uncle once got sick when he was young, and they vaccinated him, but the vaccines disabled him apparently. So, we are very skeptical and scared for my child to take any vaccines since we have knowledge of that occurrence” (PS1-1). Others indicated that they had limited information about the vaccines or had not yet seen the benefits of the vaccines, stating “thing is that we have not heard anyone that has taken the vaccine and how they felt but if we could hear them saying they felt fine after taking the vaccine, then our minds will relax”.

### 3.3. Children’s Perspectives on COVID-19 Vaccines

While COVID-19 vaccinations for children and adolescents were not available in South Africa during the data collection period, findings revealed that all the participants had heard of COVID-19 vaccines. Most were told about the vaccines by their parents or caregivers and the majority were willing to receive a COVID-19 vaccine, reflecting a great level of vaccine acceptability in the sample (see Figure 2).

Children who expressed acceptance towards COVID-19 vaccines generally referred to the benefits of the vaccines. One of the primary responses that emerged was an optimistic view that COVID-19 vaccines will “help us not to get sick” (PC2-2_F_6) and will “protect us” (PA3-2_F_13) during the pandemic. In this regard, many participants (n = 9) placed hope in COVID-19 vaccines to promote societal health and wellbeing, as shown below.

“[it will] make us better […] It will help us not to get sick and we won’t die.” (PC2-2_F_6)

“They will help people and help people not to get sick.” (PC2-3_F_6)

“It will help people and we can go back to how things are […] I think it will help and cure a lot of people who had COVID-19 and people will be happy to have the vaccine.” (PA1-2_F_13)

While a strong sense of hope in COVID-19 vaccines was evident in the children’s narratives, the narratives also suggested limited vaccine literacy, where some children misunderstood the intention of the vaccines as something that “will cure us”. While this may be interpreted as trust or hope in the vaccines’ ability to improve and restore societal health amid the pandemic, limited vaccine literacy was a standing concern in South Africa at the time.

Limited COVID-19 vaccine literacy also emerged when children were asked if they had any fears related to COVID-19 vaccines. Less than half of the participants expressed concerns about COVID-19 vaccines (n = 6), where only one child participant conveyed fears around receiving the vaccine via an injection, stating “I do not want to be pricked!” (PS1-2_M_6). For others, vaccination apprehension was shaped by a limited understanding of—or misinformation about—COVID-19 vaccines. For example, few participants described COVID-19 vaccines as something that was administered to persons who were ill and therefore they did not require vaccination. This was best reflected in the statement: “I will not take it since I am not sick” (PS2-2_F_16). Relating to vaccine misinformation, some participants feared that COVID-19 vaccines would promote COVID-19 infection and conveyed worries that “we might get sick [if we take COVID-19 vaccines]” (PC3-3_F_12). This is also reflected in the narratives below, where a few adolescents feared that the vaccines contained COVID-19 and were thus hesitant to receive a COVID-19 vaccine: 

“I am scared that it will make you sick or kill you.” (PA2-2_F_13)

“The vaccine is going to make us sick because they say they will inject us with the virus so it can fight it. I am scared of it!” (PS3-2_M_13)

“Some say it [COVID-19 vaccines] kills.” (PCH2-2_M_16)

Fears associated with COVID-19 vaccines also entailed feelings of mistrust or doubts in the effectiveness of the vaccines (n = 3). These viewpoints were only discussed by a few adolescents, who mainly questioned the long-term effects of the vaccines.

“Yes, what if it does not work? What if it protects one for just a certain time?” (PC1-4_F_17)

“I am concerned that this vaccine had become available within one year but on the other we do not have a vaccine for HIV, for TB, and for SIT… so, it has taken about 22 years without vaccine for those but the one for COVID-19 they were able to get it sooner? So, there is something scary about this vaccine for COVID. Perhaps they developed this vaccine for COVID, but they are not sure what will actually happen in a year or two.” (PS2-2_F_16)

### 3.4. Intergenerational Influence of COVID-19 Perspectives

The dyadic nature of the study further allowed us to compare parents’ perspectives on COVID-19 vaccines with that of their children. As illustrated in Table 3, amongst those children who indicated that they would receive a COVID-19 vaccine, their parents were also willing to receive a vaccine. Similarly, amongst parents who were unsure about COVID-19 vaccinations, their children expressed similar sentiments.

One example of this intergenerational transfer of COVID-19 vaccine acceptability was found in the narratives of family PCH1-1. Both mother and child indicated that they would receive a COVID-9 vaccine, explaining:

Mother: “Yes. I will [take a COVID-19 vaccine] because I trust it and that it will protect me. The treatment I already taking is protecting me from HIV/AIDS so this will too.”(PCH1-1)

Child: “It [COVID-19 vaccines] will protect us, and I heard a lot have been vaccinated around [my province]. People have to get it!”(PCH1-2)

As is evident, mother and child both referred to the protective value of COVID-19 vaccines, which justified their decisions to receive a vaccine. Another example was found in family PC1-1 below, where the mother and father participated in the study. Both parents expressed positive attitudes towards COVID-19 vaccination, stating: 

Mother: “I was happy because it [COVID-19 vaccines] would minimize the risk of people contracting the virus = provided people also do take care of themselves because this is no guarantee that when you have been vaccinated you would therefore not get infected with the virus. It is something that gives some hope when you have gotten the vaccination, like if you are using public transport, you would feel protected, and the virus will play far from you, provided you also adhere!”(PC1-1)

Father: “The fact that the vaccine is already here […] I am happy the process has started, and there is hope for us the ordinary citizens as well to get it.”(PC1-3)

Echoing the same sentiments, children in this family explained “It [COVID-19 vaccines] would help by preventing the infection” (Participant PC1-2M) and “It [COVID-19 vaccines] is coming to reduce the rate at which a number of people get infected. It is going to help us” (Participant PC1-4F). In this family, parents and children thus generally shared similar perspectives and knowledge on the vaccines, where both parties recognized that COVID-19 vaccines protected against severe COVID-19 illness.

This was also observed in narratives on COVID-19 vaccine literacy, this time related to the transfer of misinformation. In family PS2, both parent and child likened COVID-19 vaccines to something that is taken only when someone has been infected with COVID-19. The mother for example stated “I will take it [COVID-vaccines] only when I am very sick […] So, I do not see a need for us to get the vaccine while we do not know the after and before effects of it. I also feel there is no need to take it if you are not feeling sick” (Participant PS2-1). Similar misconceptions of COVID-19 vaccines were found in the child’s narrative, who explained “I think it [COVID-19 vaccines] will help many people like those with weak immune systems; like the elderly and the kids”. She further added that she is excited about the vaccines because “people will be treated for COVID-19” (PS2-2F).

## 4. Discussion

Parents generally reflected a willingness to receive a COVID-19 vaccine, which has also been found elsewhere [27,28]. Parents generally felt optimistic about the vaccine, believing that it would facilitate a “return to normal life” and that it would protect them and their families from contracting COVID-19. However, they also raised concerns about the side effects of the vaccines, questioning their safety, effectiveness, and trustworthiness. This finding corresponded with local and international studies, where fears of side effects and lack of trust in effectiveness and safety of COVID-19 vaccines have been reported [27,28,29]. Parents who were accepting of COVID-19 vaccination for children perceived vaccines as beneficial to keeping children safe from contracting the coronavirus. These findings contradicted previous studies that reported higher rates of vaccine hesitancy with regards to children receiving COVID-19 vaccination. These studies found that parents’ hesitancy was associated with limited information about the safety of COVID-19 vaccines, a lack of trust, and concerns about the side effects of COVID-19 vaccines [30,31,32].

Our study contributes to the paucity of literature exemplifying the voices of minors in discussions on COVID-19 vaccines. Vaccine acceptance was found in the children’s data, where most of the sample were hopeful about COVID-19 vaccines. Although this finding contrasted with the COVID-19 vaccination context in South Africa at the time of the study, where vaccine hesitancy was a national public health concern [8], it resonated with the global literature. COVID-19 vaccine acceptance amongst adolescent samples have been reported in China [12] and the United States [13], where vaccine acceptability has been facilitated by perspectives that vaccines would protect from COVID-19 infection and the belief that COVID-19 vaccines were safe [12]. However, more recent work by Wong and colleagues [14] in Hong Kong and Fazel and colleagues [15] in England showed lower COVID-19 vaccine acceptance amongst younger cohorts. In this regard, and given the small sample size in our study, the findings of our study should be interpreted with caution; larger surveillance studies are better able to predict COVID-19 vaccine acceptance and hesitancy among South African children and adolescents.

In addition to offering insights into parents’ and children’s perspectives and experiences during the COVID-19 pandemic, the findings also contribute to knowledge on health literacy within the context of parent–child relationships more broadly. The findings highlighted the role that vaccine literacy played in vaccination decision making. For example, limited vaccine literacy was observed in the parents’ and children’s narratives and children who expressed vaccine hesitancy also had parents with limited vaccine literacy. Papers published earlier in the pandemic called for the prioritization of critical health literacy to facilitate informed decision making pertaining to health-related behaviors and vaccination [3,8,33]. This urgency stemmed from previous research that established the value of health literacy to promote public health [33]. As pointed out by Riiser et al. [34], health literacy can impact the effective use of health knowledge.

Pertaining to parents, studies have also shown the pertinent role that parental health literacy plays in children’s health outcomes [35]. de Buhr and Tannen [36] explained that parents with young children are an important aspect of the wider population. While parents are responsible for their own life and wellbeing, they are also caregivers. They are responsible for the health and wellbeing of their children and the children are dependent on them regarding decisions related to their health. Parents’ inadequate knowledge and skills regarding health-related matters could have a negative impact on the health of children. Indeed, parental health behaviors are key variables in developing and perpetuating behaviors and attitudes among young people [37]. This was further supported by the finding that children whose parents were willing to receive a vaccine and were willing to allow their children to receive a vaccine, also expressed vaccine acceptability themselves.

Furthermore, in the current convalescent period, where COVID-19 fatalities and risks have significantly decreased in South Africa (and globally) and social interactions have largely been reestablished, societal urgencies around COVID-19 vaccination have also somewhat subsided. However, medical researchers have challenged this, given that a resurgence of COVID-19 is possible through the different variants and only 12% of people on the African continent have received one dose of a COVID-19 vaccine [38]. Mahdi [38] also calls attention to the high numbers of yearly deaths caused by seasonal influenza and tuberculosis in South Africa, further amplifying the importance of vaccinations to decrease severe health complications, particularly for high-risk groups. In all, the importance of investing in targeted strategies to promote informed health-related decision making and health literacy, including the uptake of vaccinations and vaccine literacy, is supported by our findings and the global literature [3,33,39]. Such communication strategies must pay attention to the interplay of social conditions, such as cultural, religious, and socioeconomic conditions [8], which is pivotal in a racially and culturally diverse society such as South Africa.

Finally, it would be remiss not to comment on the limitation of this study. Our sample entailed a small number of children, adolescents, and parents, who were recruited online. While the sample offered some heterogeneity in terms of family groups, further research is necessary to unpack how adolescents across different socio-cultural and racial conditions perceive COVID-19 vaccines and other vaccines and the implications of this for health-related decision making. The COVID-19 pandemic brought to the fore the importance of health knowledge, literacy, and behaviors; various lessons can be learned about how families, including adults and children, prioritize health-related decision making. Specific attention should be given to the impact that parents’ health literacy and health information-seeking behaviors has on their children’s health behaviors. This is an under-researched area of investigation in South Africa.

## Figures and Tables

**Figure 1 vaccines-11-01305-f001:**
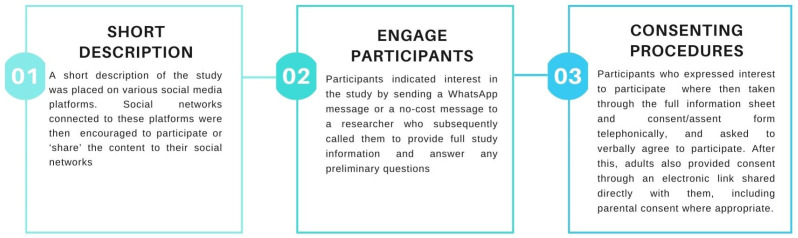
Digital snowball recruitment approach.

**Figure 2 vaccines-11-01305-f002:**
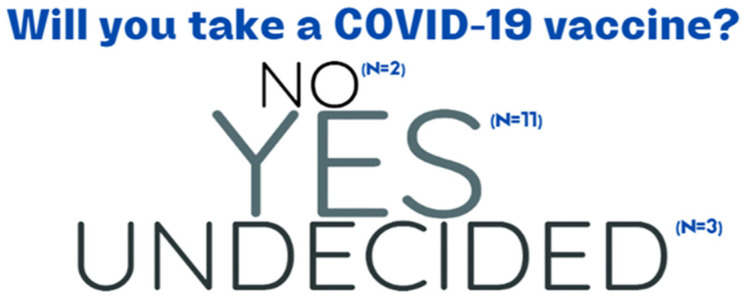
Children’s willingness to receive a COVID-19 vaccine.

**Table 1 vaccines-11-01305-t001:** Sample breakdown.

Participation Group	Male (n = 8)	Female (n = 21)	Total (*N* = 29)
Children	n = 2	n = 4	*N* = 6
Adolescents	n = 3	n = 7	*N* = 10
Adult parents/caregivers	n = 3	n = 10	*N* = 13

**Table 2 vaccines-11-01305-t002:** Participants’ characteristics.

Adult Parents/Caregivers	Adult Ages	Child Participants	Children Ages
(n = 13)		(n = 16)	
PS1-1_F	31	PS1-2_M	6
PA3-1_M	41	PA3-2_F	13
PCH1-1_F	48	PCH1-2_M	17
PCH2-1_M	19	PCH2-2_M	16
PC2-1_F		PC2-2_F	7
	37	PC2-3_F	7
		PC2-4_F	7
PC3-1_F	38	PC3-2_F	10
		PC3-3_F	12
PS2-1_F	41	PS2-2_F	16
PA2-1_F	48	PA2-2_F	13
PC1-1_F PC1-3_M	42	PC1-2_M	11
	47	PC1-4_F	17
PA1-1_F	29	PA1-2_F	13
PO1-1_F	65	PO1-2_M	13
PS3-1_F	42	PS3-2_M	13

**Table 3 vaccines-11-01305-t003:** Willingness to receive a COVID-19 vaccine.

Adult Participants	Parents’ Willingness to Receive a COVID-19 Vaccine	Parents’ Willingness for Their Child to Receive a COVID-19 Vaccine	Child Participants	Child’s Willingness to Receive a COVID-19 Vaccine
PS1-1_F	Ambivalent	Skeptical	PS1-2_M	No (“give it to those who are sick (referring to COVID-19)”)
PA3-1_M	Ambivalent	Skeptical	PA3-2_F	Yes
PC3-1_F	No (“because I don’t have COVID”)	Yes (“only if needed”)	PC3-2_F	Yes
			PC3-3_F	Unsure
PS2-1_F	Yes (“only if I have COVID”)	Yes	PS2-2_F	No (“because I am not sick (referring to COVID-19”)
PO1-1_F	Yes, but skeptical	Yes	PO1-2_M	Unsure (“only if my family says I should”)
PA2-1_F	Yes	Yes	PA2-2_F	Yes
PC1-1_F PC1-3_M	Yes	Yes	PC1-2_M	Yes, but skeptical
	Yes	Yes	PC1-4_F	Yes
PCH1-1_F	Yes	Yes	PCH1-2_M	Yes
PCH2-1_M	Yes	Not sure	PCH2-2_M	Yes
PC2-1_F	Yes	Yes	PC2-2_F	Yes
			PC2-3_F	Yes
			PC2-4_F	Yes
PA1-1_F	Yes	Yes	PA1-2_F	Not sure
PS3-1_F	Yes	Yes	PS3-2_M	Yes

## Data Availability

All relevant data are within the manuscript and Appendix A.

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
