# Peer review of "“It Doesn’t Cure, but It Protects”: COVID-19 Vaccines through the Eyes of Children and Their Parents"

_vaccines, 2023, doi:10.3390/vaccines11081305_

Round 1
Reviewer 1 Report
In the manuscript entitled “It doesn’t cure, but it protects”: COVID-19 vaccines through 2 the eyes of children and their parents.” the authors collected qualitative data via a mixed-methods study that explored South Africans’ experiences during the COVID-19 pandemic between 2020 and 2021. However, I have serious concerns about the research, mainly the methodology. I will summarize my main concerns:
1) Authors must state how the number of individuals included in the study has been calculated and the statistical model employed to establish it.
2) Authors declared “This article forms part of a longitudinal study, using a mixed methods design, that explored the impact of the COVID-19 pandemic on family life amongst a purposive sample of South African children and adults (N=45)”, but then, only N=29 was included in the study. Why?
3) Which model was employed to design the interviews to study the hesitance/acceptance of the COVID vaccines? (e.i.: The Health Belief Model (HBM), https://doi.org/10.1080/21645515.2023.2177068).
4) The authors stated that they followed the COREQ guidelines, but no COREQ checklist was included as a file or supplementary file. Please include it.
5) It is important to note that the ETHICS STATEMENT IS JUST A DRAFT. (Ethical clearance to conduct the study was provided by xxx with approval number xxx). It is mandatory to include this information in the paper.
6) The interview guide is not included (neither as a file/table nor as a supplementary file.
7) Which methodology was employed to quantitative analyse the data?
In addition, the results section is tough to read. Please, re-write it in order to make it more accessible to readers.
The quality of the English Language is adequate
Author Response
To the reviewer
Thank you for your input on our paper. Please see our responses below, along with some directions on where revisions can be found in the body of the paper.
Reviewer comment 1: Authors must state how the number of individuals included in the study has been calculated and the statistical model employed to establish it.
This paper uses a qualitative approach through purposive snowball sampling and thus does not require a statistical model to calculate the sample size. However, we have illustrated our recruitment strategy in more detail in Figure 1 on page 3.
Reviewer comment 2: Authors declared “This article forms part of a longitudinal study, using a mixed methods design, that explored the impact of the COVID-19 pandemic on family life amongst a purposive sample of South African children and adults (N=45)”, but then, only N=29 was included in the study. Why?
The full study sample entailed 45 participants (including older adults, students etc.) however, given the focus on parent-child relationships, the current paper draws on those interviews conducted with parents and children only. This has been clarified in the paper, under section 2 Materials and methods on page 2 lines 73-74.
Reviewer comment 3: Which model was employed to design the interviews to study the hesitance/acceptance of the COVID vaccines? (e.i.: The Health Belief Model (HBM), https://doi.org/10.1080/21645515.2023.2177068).
Interview guides were developed with the intention of using simple questions to capture participants’ thoughts and experiences of events as they unfolded during the COVID-19 lockdown. At the time of the study, which was in 2021, a short series of questions were used to glean the participants’ initial perspectives on the vaccines as part of the broader interview guide. See supplementary file 2. Given the exploratory nature of the study and the fluidity in our engagement approach with the participants, at the time, we adopted an inductive approach to the research questions and analyses, which we have also further explained in section 2 of the paper. Further, in the discussion section, we used the literature on vaccine hesitancy and/or acceptance to interpret and discuss our findings, focusing also on health literacy within parent-child relationships.
Reviewer comment 4: The authors stated that they followed the COREQ guidelines, but no COREQ checklist was included as a file or supplementary file. Please include it.
Included as supplementary file 3.
Reviewer comment 5: It is important to note that the ETHICS STATEMENT IS JUST A DRAFT. (Ethical clearance to conduct the study was provided by xxx with approval number xxx). It is mandatory to include this information in the paper.
This was initially removed for reviewing purposes but it has now been added on page 5, lines 199 to 200
Reviewer comment 6: The interview guide is not included (neither as a file/table nor as a supplementary file.
This has been added as a supplementary document S2.
Reviewer comment 7: Which methodology was employed to quantitative analyse the data?
This paper is purely qualitative and did not utilize a quantitative design. No quantitative data was incorporated into the paper.
We thank the reviewer again for their review of this paper and recommendations to improve the content.
Reviewer 2 Report
the topic is interesting, there are certainly points of interest that enrich the Covid literature
However, the article has a very psychological dimension and a "narrative" methodology that goes beyond my background
Since the article is mainly "narrative", it inevitably develops in qualitative terms with little data to be able to objectively analyze the topic
Author Response
Thank you to the reviewer for the comment
Reviewer 3 Report
This is a nicely written manuscript on an important issue of parental and children dyads perceptions of Covid-19 vaccines. The research is well conducted and summarized clearly. Authors used mixed methods design to collect and analyze data. Similarly to other studies authors find that mistrust, concern about vaccine safety and limited vaccine literacy are barriers to Covid vaccination. In addition, children vaccination attitudes mirrored those of their parents.
The study was well conducted but suffers from a small sample size, n=45. I am concerned that is a major limitation of this study, and as such is less informative. It is unclear how were participants identified and how they represent the South African parents and children. How was risk of bias minimized?
Second, the study measured views and perceptions in 2021 when pandemic was at its peak. Since the pandemic has been declared over, Covid vaccines are widely available, data on safety is available for millions of vaccinated and herd and vaccine immunity limits threat of this virus likely impacting views of parents and children now. Because vaccine attitudes are not fixed and can change over time, the information provided may have limited utility.
Please note that Ref #19 is incomplete.
Page 5 lines 196 and 197 have missing information . Please replace xxx with numbers.
Author Response
To the Reviewer
Thank you for your valuable insights. We have responded to your input below.
Reviewer comment 1: This is a nicely written manuscript on an important issue of parental and children dyads perceptions of Covid-19 vaccines. The research is well conducted and summarized clearly. Authors used mixed methods design to collect and analyze data. Similarly to other studies authors find that mistrust, concern about vaccine safety and limited vaccine literacy are barriers to Covid vaccination. In addition, children vaccination attitudes mirrored those of their parents.
The study was well conducted but suffers from a small sample size, n=45. I am concerned that is a major limitation of this study, and as such is less informative. It is unclear how were participants identified and how they represent the South African parents and children. How was risk of bias minimized?
Second, the study measured views and perceptions in 2021 when pandemic was at its peak. Since the pandemic has been declared over, Covid vaccines are widely available, data on safety is available for millions of vaccinated and herd and vaccine immunity limits threat of this virus likely impacting views of parents and children now. Because vaccine attitudes are not fixed and can change over time, the information provided may have limited utility.
The qualitative nature of this paper and study does limit our potential to generalise. This said, generalisability is not an aim of the qualitative paradigm. Rather our intention was to document and describe participants' lived experiences. We do, however, discuss the transferability and dependability of the study under section 2.1 of the paper as it relates to qualitative principles. Further, we discuss the limitations of the study, and the value of the work, in the second last, and last paragraphs of the discussion.
Reviewer comment 2: Please note that Ref #19 is incomplete.
This has been added.
Reviewer comment 3: Page 5 lines 196 and 197 have missing information . Please replace xxx with numbers.
This has been added and highlighted.
Thank you again for the time invested to review this paper.
Reviewer 4 Report
Manuscript ID: vaccines-2457104
The authors presented one interesting idea to develop the work entitled "“It doesn’t cure, but it protects”: COVID-19 vaccines through the eyes of children and their parents"
The manuscript needs some modifications so that it could be better than before. It would be helpful if the authors would consider the following points:
In introduction: Please explain the development and creative work. The literature review should be carefully considered.
The hypothesis of the study should be clarified at the end of the Introduction section.
Improve discussion section.
There are some sentences that need to be rephrased.
Make Figure to show the experimental design
Line 22: Change "were " to "was"
Line 79: Add "an" before " instant messaging system "
Line 118: Add "a" before " thematic analysis"
Line 137: Change "as far possible" to "as far as possible"
Line 145: Change "appropriate interpretation of findings" to "an appropriate interpretation of the findings"
Line 157: Delete " (Forero et al., 2018)"
Line 187: Change "to offer" to "offering "
Lines 187-188: Rewrite this sentence
Lines 220-221: Rewrite this sentence
Line 431: Delete " having "
Insert the correct format style for journals in the references in the text and references list.
Minor editing of English language required
Author Response
To the Reviewer
Thank you for your input on our paper. We have responded to your queries below.
Reviewer comment 1: The authors presented one interesting idea to develop the work entitled "“It doesn’t cure, but it protects”: COVID-19 vaccines through the eyes of children and their parents"
The manuscript needs some modifications so that it could be better than before. It would be helpful if the authors would consider the following points:
In introduction: Please explain the development and creative work. The literature review should be carefully considered.
It is unclear what the reviewer requires here. We are not sure what was meant by 'explain the development and creative work'. We have made some inputs to describe certain processes but we were unsure about what the reviewer was referring to here.
Reviewer comment 2: The hypothesis of the study should be clarified at the end of the Introduction section.
This paper reports qualitative data and does not need to state hypotheses. We were not testing hypotheses nor were we assessing relationships between variables. We were describing participants' lived experiences through the qualitative paradigm.
Reviewer comment 3: Improve discussion section.
There are some sentences that need to be rephrased.
Grammatical errors have been attended to.
Reviewer comment 4: Make Figure to show the experimental design
This is a qualitative, purposive paper and does not employ an experimental design.
Reviewer comments 5:
Line 22: Change "were " to "was"
Line 79: Add "an" before " instant messaging system "
Line 118: Add "a" before " thematic analysis"
Line 137: Change "as far possible" to "as far as possible"
Line 145: Change "appropriate interpretation of findings" to "an appropriate interpretation of the findings"
Line 157: Delete " (Forero et al., 2018)"
Line 187: Change "to offer" to "offering "
Lines 187-188: Rewrite this sentence
These edits have been made.
Lines 220-221: Rewrite this sentence
This cannot be rephrased as it entails the participant’s voice.
Line 431: Delete " having "
Done
Thank you again for reviewing this paper. We hope the changes that have been made have improved the paper.
Round 2
Reviewer 1 Report
The authors have addressed my concerns and included my suggestions.